# RNA Splicing Defects in Hypertrophic Cardiomyopathy: Implications for Diagnosis and Therapy

**DOI:** 10.3390/ijms21041329

**Published:** 2020-02-16

**Authors:** Marta Ribeiro, Marta Furtado, Sandra Martins, Teresa Carvalho, Maria Carmo-Fonseca

**Affiliations:** 1Instituto de Medicina Molecular João Lobo Antunes, Faculdade de Medicina, Universidade de Lisboa, Av Prof Egas Moniz, Edificio Egas Moniz, 1649-028 Lisboa, Portugal; martaribeiro@medicina.ulisboa.pt (M.R.); martafurtado@medicina.ulisboa.pt (M.F.); sandramartins@medicina.ulisboa.pt (S.M.); t.carvalho@medicina.ulisboa.pt (T.C.); 2Department of Bioengineering and iBB–Institute for Bioengineering and Biosciences, Instituto Superior Técnico, Universidade de Lisboa, Av. Rovisco Pais, 1049-001 Lisboa, Portugal

**Keywords:** hypertrophic cardiomyopathy, RNA splicing, splicing mutations, RNA genetic testing, RNA therapeutics

## Abstract

Hypertrophic cardiomyopathy (HCM), the most common inherited heart disease, is predominantly caused by mutations in genes that encode sarcomere-associated proteins. Effective gene-based diagnosis is critical for the accurate clinical management of patients and their family members. However, the introduction of high-throughput DNA sequencing approaches for clinical diagnostics has vastly expanded the number of variants of uncertain significance, leading to many inconclusive results that limit the clinical utility of genetic testing. More recently, developments in RNA analysis have been improving diagnostic outcomes by identifying new variants that interfere with splicing. This review summarizes recent discoveries of RNA mis-splicing in HCM and provides an overview of research that aims to apply the concept of RNA therapeutics to HCM.

## 1. Introduction

Familial hypertrophic cardiomyopathy (HCM), the most common inherited heart disease, is predominantly caused by mutations in genes that encode proteins associated with the cardiac sarcomere. HCM has an estimated prevalence ranging between 1:200 and 1:500 individuals in the general adult population [1,2] and is best known for causing sudden cardiac death, particularly in young athletes. Approximately two-thirds of patients with HCM are asymptomatic or minimally symptomatic [3,4,5], and the most frequent symptoms include exertional dyspnea, chest pain, and arrhythmias [6,7]. These symptoms can be relentlessly progressive with resultant heart failure, underscoring the importance of therapeutic strategies to slow, halt, or reverse disease progression [6,7]. Current first-line medical treatment aims to alleviate the symptoms but does not specifically target the pathophysiology of the disease. Patients that remain symptomatic despite pharmacological treatment are often referred for septal reduction therapies (surgical myectomy or percutaneous alcohol septal ablation), and for refractory patients with advanced heart failure, implantation of a left ventricular device or cardiac transplantation may be the only option. Thus, new therapies that precisely modify HCM pathophysiology are critically needed. To encourage research in this field, HCM has recently been given orphan disease status [8]. 

The clinical hallmark of HCM is left ventricular hypertrophy (defined by an end-diastolic ventricular septal thickness in adults > 15mm) occurring in the absence of abnormal loading conditions or other secondary causes [9]. In histopathologic examinations, HCM is characterized by enlarged cardiomyocytes (which can be 20–30 µm in diameter, in contrast to a diameter of 5–12 µm in normal cells) with bizarre shapes, distorted nuclei, and a loss of normal parallel alignment, giving a disarrayed appearance to the myocardial architecture intermingled with increased extracellular fibrosis [10].

Genetic testing is a valuable tool in confirming the diagnosis of HCM. Indeed, the identification of a mutation that is causative of HCM will support diagnosis in a proband. In such cases, cascade screening is recommended, that is, testing for the presence of the mutation in family members. This type of family screening helps to identify high-risk individuals before the occurrence of overt symptoms and reassure those with a negative test. Mutation carriers should be counseled appropriately, and prenatal or preimplantation tests may be considered. Genetic analysis is also useful to confirm the diagnosis of HCM in ambiguous situations. For example, because HCM is the most common cause of sudden cardiac death in young athletes, who often exhibit left ventricular wall thicknesses between 13 and 18 mm [11], it is critical to distinguish the physiological hypertrophy of athletes from the pathological hypertrophy of HCM. On the other hand, a positive test supports an HCM diagnosis in individuals with a septal thickness below the cutoff point for clinical diagnosis [12,13]. Another important contribution of genetic testing is that it allows for a distinction between HCM and so-called phenocopy conditions, i.e., apparently similar disorders with different causes. Namely, this includes autosomal dominant cardiomyopathies caused by *PRKAG2* mutations, X-linked cardiomyopathies such as Fabry’s disease and Danon’s disease, glycogen storage diseases, lysosomal storage diseases, mitochondrial diseases, and triplet repeat syndromes, all of which share clinical features with sarcomeric hypertrophic cardiomyopathy yet are distinct disorders with different natural histories and treatments [7].

To date, more than 1500 mutations have been reported to be associated with HCM, but due to the rarity of individual mutations, the available clinical data are insufficient to support meaningful genotype–phenotype correlations. Thus, for the majority of HCM patients, a positive genetic test is unable to predict the clinical course of the disease or the risk of complications, including sudden cardiac death and heart failure. A subset of HCM patients (~5%) have two or more mutations, either in the same gene or in different genes, and these compound or double heterozygosity genotypes are often associated with particularly severe disease progression [7].

A major limitation of current genetic testing is its failure to identify a causative mutation in 50%–60% of HCM patients [14]. Initially it was thought that patients with a negative test had mutations in “missing” genes not yet associated with HCM. Although the full spectrum of HCM genes was expected to be quickly determined by using high-throughput DNA sequencing approaches, which allow for whole-exome and whole-genome analyses, the accumulated results did not improve the effectiveness of HCM gene-based diagnoses. Rather, access to high-throughput DNA sequencing data has vastly expanded the number of variants of uncertain significance, leading to many inconclusive results that limit the clinical utility of genetic testing [15]. More recently, developments in RNA analysis have been improving diagnostic outcomes by identifying new variants that interfere with splicing. Here, we provide an overview of RNA splicing mechanisms, and we discuss how mutations that interfere with splicing cause disease. Next, we focus on HCM-associated splicing mutations, giving special emphasis to recently discovered noncanonical splicing mutations. Finally, we discuss emerging strategies for HCM-targeted RNA therapeutics.

## 2. RNA Splicing Mechanisms

The vast majority of human protein-coding genes consist of multiple short pieces of coding sequences termed exons, which are separated by much longer intervening noncoding sequences or introns that are removed by splicing (Figure 1). This gene architecture, which is made up of an alternation of multiple exons and introns, sets the basis for alternative splicing, allowing for the generation of multiple mRNA isoforms from a single gene. Alternative splicing has been estimated to occur in nearly all human multi-exonic genes, producing mRNA splice variants that tend to be differentially expressed in a cell- and tissue-specific manner [16,17]. Alternatively spliced mRNA isoforms may encode proteins with distinct functions or may affect the expression, localization, and/or stability of mRNA [18]. Regulated switches in splice site selection trigger the expression of specific isoforms during development and cell differentiation or in response to external stimuli [19]. For example, alternative splicing plays an important role in cardiac remodeling during development, particularly as the fetal heart adapts to birth and converts to an adult function [20].

During transcription, the entire sequence information of a gene is copied into a precursor messenger RNA molecule (pre-mRNA). Pre-mRNA splicing consists of excising the introns and ligating the flanking exons (Figure 1A,B). The splicing reaction is catalyzed by a highly sophisticated ribonucleoprotein machinery called the spliceosome, which is formed by five small nuclear RNAs (the U1, U2, U4, U5, and U6 snRNAs) and more than 200 proteins [21]. A subset of these proteins is associated with snRNAs forming functional ribonucleoprotein particles, termed snRNPs. Each snRNP corresponds to a complex between an snRNA molecule, a common set of proteins (termed Sm proteins), and a variable number of additional specific proteins [22,23].

Spliceosomes form anew on every intron, recruited in part through base-pairing interactions between the snRNAs and four short conserved sequences that define the exon–intron boundaries in pre-mRNA (Figure 1A). These sequence elements are the 5’ splice site (located at the start of the intron), the 3’ splice site (located at the end of the intron), the branch point sequence (located upstream of the 3’ splice site), and the polypyrimidine tract (located between the branch point and the 3’ splice site). Spliceosome assembly starts with the recognition of the 5’ss by U1 snRNP (Figure 2A), followed by base-pairing between U2 snRNA and pre-mRNA sequences adjacent to the 3’ splice site. However, the latter interaction requires prior binding of a protein called splicing factor 1 (SF1) to the branch point sequence; additionally, the 65-kDa and 35-kDa subunits of the U2 snRNP auxiliary factor (U2AF65 and U2AF35) must bind, respectively, to the polypyrimidine tract and the conserved AG dinucleotide at the intron’s 3’ end (Figure 2A). Preassembly of this complex is crucial for the subsequent recruitment of the U2 snRNP, which displaces SF1 and binds to BP (Figure 2B). Next, interactions are formed between the U1 and U2 snRNPs, bringing the 5’ and 3’ splice sites in close spatial proximity. The stable binding of the U2 snRNP to pre-mRNA triggers the subsequent recruitment of the preassembled U4/U6.U5 tri-snRNP (Figure 2C). With all snRNPs present, the spliceosome undergoes major conformational rearrangements that involve the release of U1 and U4 and the base pairing between the U6 and U2 snRNAs (Figure 2D): these interactions trigger the activation of the catalytic center. At this stage, the spliceosome initiates splicing, which consists of two transesterification reactions (i.e., the replacement of one phosphodiester linkage with another reaction). In the first reaction, the 2’-hydroxyl group of the branch point adenosine carries out a nucleophilic attack on the phosphate group at the 5’ splice site phosphodiester bond, generating a free 5’ exon and an intron lariat-3’ exon intermediate. The second transesterification reaction occurs when the exposed 3’-hydroxyl of the 5’ exon attacks the phosphodiester bond at the 3’ splice site, cleaving the lariat-structured intron and ligating the two exons (Figure 1B). The U5 snRNA interacts with both exons and is critical to aligning the exons for ligation in the second catalytic reaction, during which there is a 5’ss-mediated attack on the 3’ss, giving rise to the spliced product and releasing the intron lariat (Figure 2D). Upon the completion of splicing, ligated exons are released from the spliceosome, and the remaining snRNPs persist, transiently associated with the intron lariat (Figure 2E). Subsequently, the spliceosome disassembles and the excised intron is degraded (Figure 2F). 

The vast majority of introns are recognized and removed by the U1, U2, U4, U5, and U6 snRNP complex, also known as the major spliceosome. However, a minority of introns are spliced by a distinct type of snRNP complex called the minor spliceosome [24]. Overall, the major and minor spliceosomes share many common features, and the mechanism of splicing is nearly identical. However, the minor spliceosome is composed of four distinct snRNAs termed U11, U12, U4atac, and U6atac, which have a counterpart in U1, U2, U4, and U6, respectively [25].

Splice site recognition by spliceosome components is often facilitated by additional proteins that recognize specific sequences in exons and introns (called splicing enhancers) (Figure 3). Namely, the binding of serine/arginine (SR)-rich splicing factors to exonic and intronic splicing enhancers helps stabilize the interaction between the U1 and U2 snRNPs and the 5’ and 3’ splice sites [26]. There are also pre-mRNA sequence elements located in exons or introns that inhibit splice site recognition and are therefore called splicing silencers (Figure 3). Most splicing silencers function as binding sites for members of the heterogeneous nuclear ribonucleoprotein (hnRNP) family [27].

Depending on the combinatorial effect resulting from the binding of proteins that either promote or repress the recognition of conserved splicing sequences by the spliceosome, some exon–intron boundaries in pre-mRNA can be differentially selected, leading to the production of alternatively spliced mRNA [28]. Exons that are always included in mRNA are called constitutive exons, and exons that are sometimes included and sometimes excluded from mRNA are called cassette exons. Some pre-mRNA contains multiple cassette exons that are mutually exclusive, producing mRNA that always includes one of several possible exon options. Exons can also be lengthened or shortened by altering the position of one of their splice sites (alternative 5’ and alternative 3’ splice site selection), and certain intronic sequences may persist in the final mRNA (intron retention). Finally, the first and last exons can differ, depending on the use of alternative promoters or alternative polyadenylation sites, respectively. The control of alternative splicing decisions involves multiple mechanisms, including RNA-binding proteins that interact with pre-mRNA and modulate the efficiency of splice site recognition by the spliceosome, the formation of secondary structures in RNA, the transcription rate, and the epigenetic modification of the template chromatin [28,29]. An increasing number of factors with a role in splicing regulation have been identified [30]. Some are tissue-specific RNA-binding proteins, while many others are ubiquitously expressed, but their relative abundances can fluctuate in different tissues [31].

## 3. Disease-Causing Splicing Mutations 

There are diverse mechanisms by which mutation-induced defects in RNA splicing act as a primary cause of disease [32,33,34].

One category of splicing mutations includes single-nucleotide substitutions that disrupt constitutive or alternative splice sites. The two highly conserved positions at the 5’ and 3’ splice junction (typically GT and AG, respectively; see Figure 1A) are called the “essential” or “canonical” splice site nucleotides, and substitutions at these positions in haplo-insufficient autosomal dominant genes are diagnostically classified as pathogenic. However, recognition of a splice site by the spliceosome relies on a larger “splice region” composed of less well-conserved sequences (Figure 1A). Although several studies have highlighted the importance of the near-splice-site region [35,36,37], variants in this region are often diagnostically classified as variants of unknown significance (VUS).

Splice site mutations result in either exon skipping, the activation of a nearby cryptic or alternative 3’ or 5’ splice site, or intron retention (Figure 4A). The use of unnatural splice sites or intron retention frequently changes the reading frame for translation and introduces premature termination codons (PTCs) into mRNA (Figure 4B). Similarly, skipping some exons alters the reading frame and introduces PTCs. In cells, most abnormal mRNA-harboring PTCs are recognized and degraded by a quality-control mechanism termed nonsense-mediated mRNA decay (NMD) [38]. The degradation of PTC-containing mRNA by NMD prevents the translation of truncated proteins with potentially deleterious gain-of-function or dominantly negative activity. Thus, in most cases, splice site mutations result in the inactivation (loss of function) of the mutated allele. However, some mRNA-containing PTCs escape NMD and are translated into truncated proteins. If the skipped exon does not alter the reading frame, then a shorter protein will be synthesized. In other cases, the mutations induce the expression of alternatively spliced protein isoforms with different functional properties: rather than create an aberrant (cryptic) splice site usually associated with a loss of function, these mutations shift the ratio of expression of natural protein isoforms. 

A second category of mutations disrupts exonic splicing regulatory elements (Figure 3). Remarkably, the majority of human exons are predicted to contain exonic splicing elements [39,40], suggesting that many disease-causing exonic mutations classified as synonymous, missense, or nonsense could be unrecognized splicing mutations [41]. Exonic mutations that affect RNA splicing may induce skipping of the mutant exons, leading to either the synthesis of a shorter protein or the inactivation of mutant allele expression. 

Mutations in the third category are located in internal regions of introns (deep-intronic mutations). For a long time, medical genetic testing has focused mainly on sequencing the exons and the exon–intron boundaries, searching for mutations that are more likely to affect the function of the encoded protein. However, in recent years, advances in whole-genome sequencing techniques have resulted in the identification of an increasing number of disease-causing mutations located more than 100 nucleotides away from splice junctions [42]. Deep intronic mutations most commonly lead to pseudo-exon inclusion due to the creation and activation of cryptic splice sites (Figure 4A). Alternatively, deep intronic mutations disrupt splicing regulatory elements located within introns. Deep intronic mutations can also interfere with the function of transcription regulatory motifs and noncoding RNA genes [33,42]. 

In contrast to the previous categories, which all refer to mutations that affect the expression of a single gene by disrupting a splicing *cis*-element, mutations in the fourth category have a *trans* effect on multiple genes by interfering with the function of either core components of the spliceosome or splicing regulatory proteins. For example, some patients with retinitis pigmentosa have mutations in genes that encode for protein components of the U4/U6.U5 tri-snRNP [43,44,45]. Another example is spinal muscular atrophy, one of the most common genetic causes of childhood mortality, which is caused by loss-of-function mutations in a gene encoding a protein required for the assembly of spliceosomal snRNPs [46]. More recently, mutations in genes encoding for either protein or snRNA components of the minor spliceosome have been identified as causative of several diseases [25].

## 4. HCM-Associated Splicing Mutations 

Typically, HCM is inherited in an autosomal-dominant manner, with incomplete penetrance and high phenotypic variability, even within the same family [47,48]. The proportion of mutation carriers with clinically detectable disease tends to increase with age, and in most persons, hypertrophy is manifested in adolescence [48]. However, due to the incomplete penetrance, patients with subtle HCM features are difficult to identify in the general population.

HCM can be caused by a single allelic mutation in any of at least eight genes that encode for cardiac sarcomere-associated proteins [15,49,50]. These are the beta-myosin heavy chain (*MYH7*), cardiac myosin-binding protein C (*MYBPC3*), cardiac troponin T (*TNNT2*), cardiac troponin I (*TNNI3*), cardiac alpha-actin (*ACTC*), alpha-tropomyosin (*TPM1*), myosin ventricular essential light chain 1 (*MYL3*), and myosin ventricular regulatory light chain 2 (*MYL2*) genes. Mutations in *MYH7* and *MYPBC3* occur the most often, and together they account for approximately half of all HCM cases [15,50].

The vast majority of HCM mutations in *MYH7*, as well as most other sarcomere mutations (Figure 5), have been classified as missense, i.e., they are point mutations that result in the substitution of one amino acid for another in a protein [15]. Studies have found normal levels of the mutant MYH7 protein, but its function is perturbed. Specifically, biophysical properties of myosins that contain HCM mutations indicate a gain of function, with enhanced myosin ATPase activity, increased generated force, and accelerated actin filament sliding [51]. In contrast, most *MYBPC3* mutations have been classified as nonsense, frameshift, or splice site mutations (Figure 5). A nonsense mutation is a single nucleotide substitution that results in a PTC in the transcribed mRNA, while a frameshift mutation can be an insertion, deletion, or duplication of nucleotides that changes the reading frame of the mRNA and often leads to a PTC. Splice site mutations may also disrupt the normal reading frame and introduce a PTC into mRNA (Figure 4B). It is therefore expected that the most frequent *MYBPC3* mutations result in the degradation of PTC-containing mutant mRNA. Indeed, an analysis of myosin-binding protein C expressed in human myectomy samples from HCM patients with *MYBPC3* gene mutations revealed the absence of truncated proteins and reduced levels of the normal protein [52]. Moreover, HCM mutations in the *MYBPC3* gene (engineered into mice) resulted in the reduced expression of myosin-binding protein C and caused altered cardiac function [53]. These results suggest that *MYBPC3* haplo-insufficiency (i.e., a reduction in the amount of normal protein due to the inactivation of the expression of the mutant allele) is a pathologic mechanism for HCM. 

As expected, the majority of HCM-associated splice site variants reported in the ClinVar database are located in the two most highly conserved nucleotide positions at the splice junction (Figure 6). Although several computational algorithms have been developed to predict the effects of single-nucleotide variants on splicing [37,54,55,56,57,58,59], the definitive test of whether a disease-causing mutation affects splicing is through the direct analysis of mRNA. Indeed, it is essential to sequence the mutant mRNA to define its splicing pattern. Moreover, the total levels of mRNA should be measured to determine whether the mutation triggers NMD and therefore reduces the expression of the mutant transcript.

Aberrant splicing caused by HCM mutations has been shown using minigenes transfected into HEK293 cells [60]. However, this is not an ideal system, because mechanisms controlling splicing decisions are known to be influenced by chromatin structure and to be cell-type-specific. Thus, RNA from the affected tissue should be studied. One way of doing this has been to analyze mRNA from left ventricular septum samples from patients undergoing septal myectomy to relieve left ventricular outflow tract obstruction [52,61,62,63]. Alternatively, RNA can be studied in cardiomyocytes differentiated in vitro from induced pluripotent stem cells (iPSCs) derived either from patients or from normal iPSC gene-edited to harbor a patient-specific genetic variant [64,65,66]. RNA analysis can also be performed on peripheral blood samples, at least for *MYBPC3* mutations. Indeed, it was recently shown that RNA extracted from fresh venous blood supports the amplification of *MYBPC3* transcripts and replicates the splicing patterns found in myocardial tissue [63]. 

RNA analyses of families with *MYBPC3* variants in the near-splice-site region allowed for a reclassification from “uncertain significance” to “likely pathogenic” [63]. Multiple variants located in the near-splice-site region and predicted to disrupt splicing by computational algorithms were shown to either activate cryptic splice sites or induce exon skipping (Figure 7). Some variants resulted in a frameshift and introduction of a PTC, while other variants resulted in shorter mRNA with in-frame deletions. RNA analyses further revealed that exonic variants classified as missense but predicted to disrupt splicing by computational algorithms caused exon skipping, leading to frameshifts [63]. Another study used whole-genome sequencing and identified deep intronic variants that resulted in the inclusion of pseudo-exons (Figure 7), leading to frameshifts [62]. A synonymous exonic variant was further shown to create a novel cryptic splice sequence that truncated the exon (Figure 7), leading to an in-frame deletion [62].

## 5. RNA Therapeutics for HCM

Besides improving diagnostics, understanding precisely the expression of relevant RNA in patient cells has the potential to inspire the development of new strategies to specifically treat an individual. Although for years the field of RNA therapeutics had to overcome numerous difficulties in achieving efficacious results without toxic side effects, the first RNA-targeted therapies have recently reached clinics, and many are advancing to the final phases of clinical trials. In 2016, the antisense oligonucleotides Eteplirsen™ and Nusinersen™ were approved for treatment of Duchenne muscular dystrophy (DMD) [67] and spinal muscular atrophy [68], respectively. These oligonucleotides form base-pair interactions with nascent pre-mRNA and alter its splicing pattern. Eteplirsen hybridizes to a site within exon 51 of the *DMD* pre-mRNA, thereby sterically blocking spliceosome assembly at that site: this results in the skipping of exon 51 and the correction of the disease-causing frameshift mutation. The corrected mRNA contains exon 50 ligated to exon 52 and generates a shortened but still functional version of the dystrophin protein. Nusinersen hybridizes to an intronic region upstream of exon 7 of the *SMN2* pre-mRNA and blocks an inhibitory signal located at that site, causing the inclusion of exon 7. The *SMN2* mRNA with exon 7 encodes a fully functional protein that substitutes for the missing SMN1 protein.

In principle, splice-switching antisense oligonucleotides could be valuable for HCM treatment in cases of disease caused by mutations that disrupt normal splicing. A proof-of-concept study has demonstrated the feasibility and efficacy of inducing the skipping of a mutated *MYBPC3* exon 6 using the viral-mediated transfer of antisense oligonucleotides in a mouse model of HCM [69]. The transduction of cardiac myocytes or the systemic administration of oligonucleotides reduced aberrantly spliced mRNA, abolished cardiac dysfunction, and prevented left ventricular hypertrophy in newborn mice [69]. Although they have not yet been experimentally tested, antisense oligonucleotides designed to hybridize to cryptic splice sites in mutant pre-mRNA could sterically block spliceosome assembly at that site, thus preventing mis-splicing. 

Another therapeutic strategy is to induce the silencing of disease-causing genes through RNA interference (RNAi) [70]. After the discovery that double-stranded RNA (often called small interfering RNA, or siRNA) can silence the expression of proteins encoded by complementary mRNA in the nematode of *Caenorhabditis elegans* [71,72], synthetic exogenous siRNA was shown to induce sequence-specific gene expression knock-down in mammalian cells [73]. RNAi is a fundamental process initiated by the presence of long double-stranded RNA that is cleaved by the enzyme Dicer into shorter fragments of 21–23 nucleotides containing two single-stranded nucleotides at their 3’ ends. Synthetic siRNA is designed to mimic the natural products of Dicer. Each siRNA comprises a sense “passenger” RNA strand and a paired antisense, or “guide”, RNA strand. The siRNA molecules are loaded onto the RNA-induced silencing complex (RISC), which is composed of Dicer and Argonaute 2 (Ago2). During RISC assembly, the siRNA is unwound, the “passenger” strand is removed, and the single-stranded antisense guide base-pairs with the mRNA target. The mRNA hybridized to the siRNA is then cleaved by Ago2, which contains an RNase H-like domain that functions to cleave one strand of the RNA/RNA duplex. Because siRNA can in principle downregulate any human mRNA, it should be ideal for eradicating the expression of disease-causing mutant alleles. However, despite major efforts, siRNA-based therapies have faced multiple hurdles. Namely, delivery and stability have proven difficult. Further, siRNA has been found to trigger innate toll-like immune receptors to initiate inflammation, raising concerns about safety [70]. Tremendous progress in the field culminated in 2018 with the first-ever siRNA product (Patisiran) approved as a therapy for the rare hereditary disease transthyretin-mediated amyloidosis [74]. More recently, siRNA-targeting mRNA isoforms responsible for the expression of the placenta-derived mediators of preeclampsia succeeded in suppressing clinical signs in a primate model [75].

In the case of HCM, selective reduction in the expression of mutant alleles that code for dominant negative proteins would be the most direct therapeutic approach. Proof-of-concept has been established in a mouse model of HCM that is heterozygous for the R403Q mutation in *Myh6* [76]. RNAi constructs delivered by an adeno-associated virus vector preferentially reduced the levels of mutant transcripts and suppressed myocardial hypertrophy and fibrosis [76]. Although siRNA can distinguish between mRNAs that differ by one single nucleotide, transcripts with splicing mutations that lead to in-frame deletions or insertions should be easier to target by RNAi. However, before pursuing any RNAi-based strategy for HCM, it is critical to ensure that the expression of the normal allele is sufficient to support normal myocardial function.

An alternative modality for the correction of mutant mRNA, with potential applications for HCM, is spliceosome-mediated RNA *trans*-splicing, or SMaRT [77]. Inspired by the observation that spliceosomes can ligate exons from two distinct pre-mRNAs, creating a chimeric mRNA, the SMaRT technology makes use of an exogenous RNA sequence to replace one or several exons of the target mutant pre-mRNA. An artificial pre-mRNA is engineered to contain the coding sequence of substitution next to an intron. The end of this artificial intron consists of a stretch of nucleotides that base-pair with the target intron in the endogenous pre-mRNA, bringing the exogenous exon in close proximity to its endogenous mutant counterpart. Efficient substitution of the mutant exon requires that the engineered *trans*-splicing out-competes the physiological *cis*-splicing process, and this currently remains a major challenge. If successful, the SMaRT technology would be particularly appealing in treating HCM, because a singly engineered RNA construct could be used to repair numerous mutations. Namely, two distinct RNA constructs covering the first and second half of the *MYBPC3* mRNA should in principle be able to repair all of the mutations in this gene and therefore treat 40%–60% of all HCM patients. Proof-of-concept studies have been reported using artificial pre-mRNA that carried the wild-type *MYBPC3* cDNA sequence from exon 1 to exon 21 juxtaposed with an intron with a 120-nucleotide binding domain for base pairing with a complementary sequence in intron 21 of the endogenous *MYBPC3* mRNA [78,79]. However, chimeric molecules resulting from *trans*-splicing represented less than 1% of all *MYBPC3* mRNAs in cardiomyocytes, indicating that the efficiency of this approach was too low to be considered as a therapeutic option [79].

Noncoding RNAs, particularly microRNAs (miRNAs), are also attracting much attention as biomarkers of cardiac disease and potential therapeutic targets. Altered expression levels of circulating miRNAs have been reported in association with hypertrophic cardiomyopathy [80,81,82,83], and the forced overexpression of stress-inducible miRNAs was shown to induce cardiomyocyte hypertrophy [84]. However, it remains to be established whether the modulation of miRNA levels is sufficient to revert the HCM phenotype. 

## 6. Concluding Remarks

Advances in high-throughput sequence technologies and computational algorithms for data analysis have expanded our ability to identify the genetic causes of HCM. Whole-genome sequencing is valuable in detecting variants in intronic regions, and new in silico tools are able to predict which variants located in introns, exons, or splice regions are more likely to alter splicing. However, RNA analysis is essential to demonstrate the consequences of potential splice-disrupting mutations. Recent studies have succeeded in reclassifying variants from uncertain significance to likely pathogenic through an analysis of RNA isolated from fresh venous blood, from myectomy samples, or from induced pluripotent stem cell-derived cardiomyocytes from patients. Besides improving the precision of genetic diagnostics, the discovery of disease-causing aberrantly spliced mRNA in HCM patients opens new venues for the development of RNA-targeted therapies. Splice-switching antisense oligonucleotides and short interfering RNAs are particularly promising strategies. Although further developments are needed to overcome major challenges related to safety and delivery, RNA-targeting drugs hold great potential to treat HCM. Indeed, hypertrophied cardiomyocytes can undergo remodeling, and the disease does not involve cell death. Moreover, the cellular perturbations caused by the mutations are subtle and tolerated until a tipping point that triggers decompensation. Thus, it may be sufficient to slightly reduce the expression of aberrant mRNAs to sustain the compensated state, particularly if therapy begins in asymptomatic mutation carriers. 

## Figures and Tables

**Figure 1 ijms-21-01329-f001:**
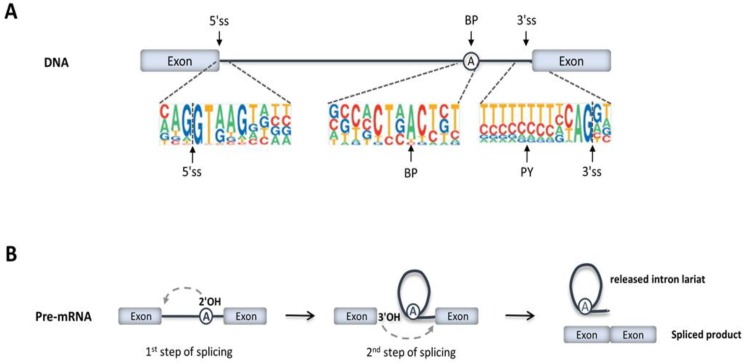
Precursor messenger RNA (pre-mRNA) splicing. (**A**) Exons are represented by boxes and introns by lines. The most conserved nucleotides at the 5’ splice site (5’ss), branch point (BP), polypyrimidine tract (PY), and 3’ splice site (3’ss) are indicated. (**B**) The two transesterification reactions that result in the excision of introns from pre-mRNA are represented.

**Figure 2 ijms-21-01329-f002:**
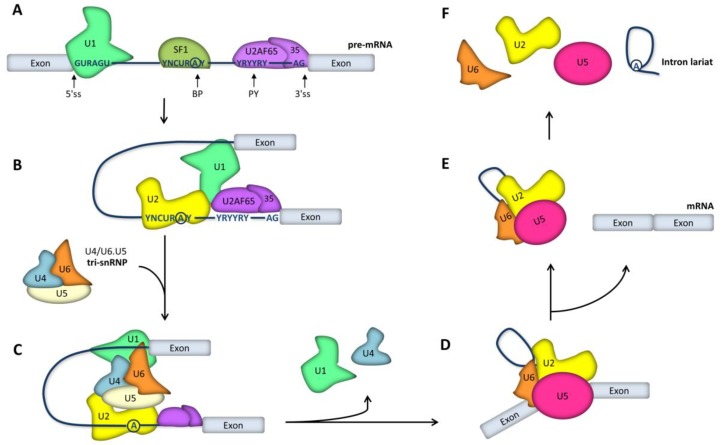
Spliceosome assembly for pre-mRNA splicing. (**A**) Spliceosome assembly is initiated through binding of the U1 snRNP to the 5’ splice site (ss), SF1 to the branch point (BP), U2AF65 to the polypyrimidine tract (PY), and U2AF35 to the 3’ splice site. (**B**) Next, the U2 snRNP displaces SF1 and binds the BP. (**C**) Recruitment of the U4/U6.U5 tri-snRNP. (**D**) Conformational rearrangements involving the release of the U1 and U4 snRNPs lead to catalytic activation. (**E**) Ligated exons are released from the spliceosome, and the remaining snRNPs persist, associated with the intron lariat. (**F**) Subsequently, snRNPs dissociate from the excised intron.

**Figure 3 ijms-21-01329-f003:**
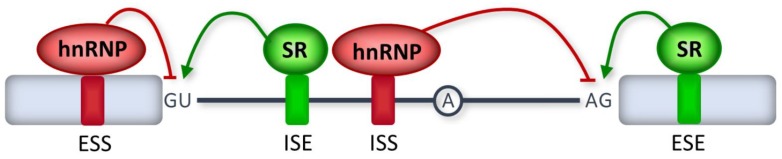
Splicing regulatory sequence elements. Regulatory elements are located in exons (exonic splicing enhancer (ESE) or exonic splicing silencer (ESS)) and in introns (intronic splicing enhancer (ISE) or intronic splicing silencer (ISS)). Enhancers (green) bind SR proteins and promote spliceosome assembly at nearby splice sites. Silencers (red) bind members of the heterogeneous nuclear ribonucleoprotein (hnRNP) family and inhibit spliceosome assembly.

**Figure 4 ijms-21-01329-f004:**
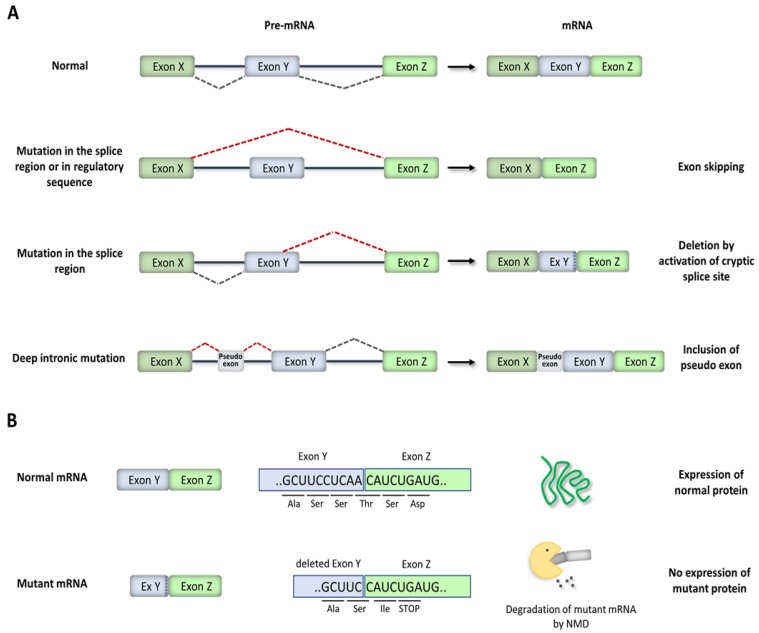
Splicing mutations. (**A**) Schematics illustrating the consequences of *cis*-acting mutations on pre-mRNA splicing. (**B**) Representation of a frameshift deletion introduced by the activation of a cryptic splice site within an exon and degradation of the resulting mRNA by nonsense-mediated decay (NMD).

**Figure 5 ijms-21-01329-f005:**
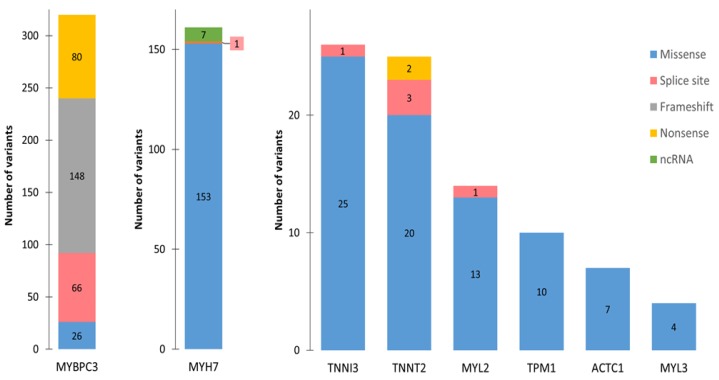
Hypertrophic cardiomyopathy (HCM) mutations. The bar graphs depict the number and type of HCM-associated variants in the indicated genes. All variants reported as pathogenic or likely pathogenic in the National Center for Biotechnology Information’s ClinVar database (http://www.ncbi.nlm.nih.gov/clinvar/) were considered.

**Figure 6 ijms-21-01329-f006:**
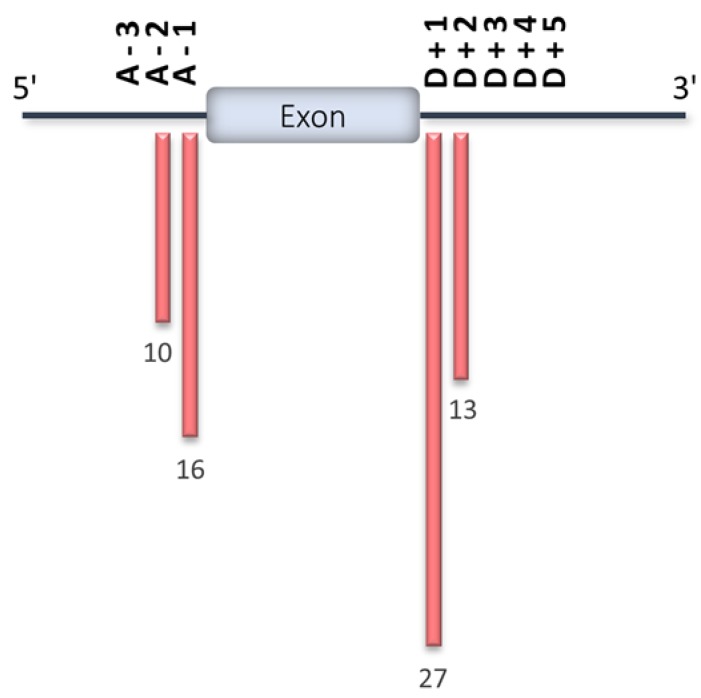
HCM splice site mutations reported in the ClinVar database. Position of splice site mutations reported in the ClinVar database for the eight HCM-associated genes shown in Figure 5. The number of variants per position is indicated. The 5′ splice site (ss) is also known as the donor (D) splice site, whereas the 3’ site is also known as the acceptor (A) splice site. Position D + 1 corresponds to the first intronic nucleotide downstream of the depicted exon, D + 2 to the second, and so forth. Position A-1 corresponds to the last intronic nucleotide upstream of the depicted exon, A-2 to the previous one, and so forth.

**Figure 7 ijms-21-01329-f007:**
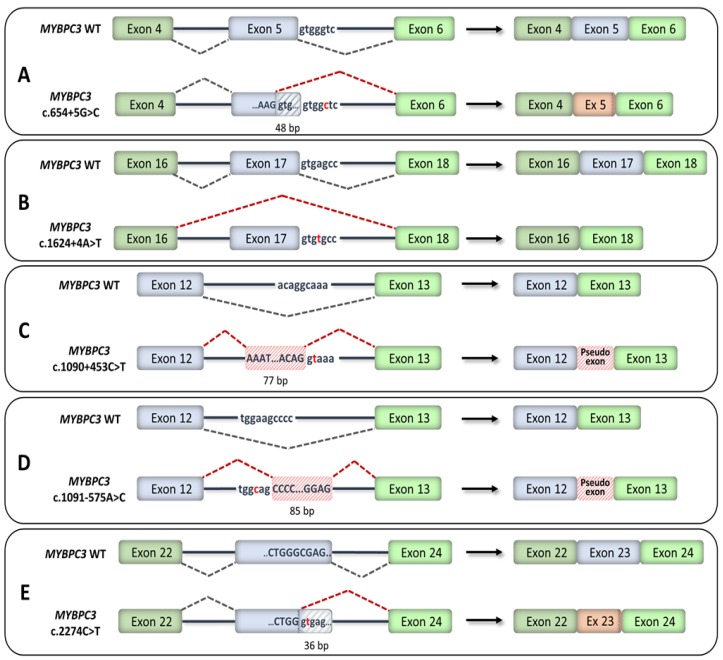
Recently reclassified *MYBPC3* splicing mutations. Exons are represented by boxes and introns by lines. Exonic nucleotides are indicated by capital letters and intronic nucleotides by small-case letters. (**A**) Variant c.654+5G>C created a cryptic donor splice site within exon 5, leading to a truncated exon with an in-frame deletion of 48 pb [63]. (**B**) Variant c.1624+4A>T caused the skipping of exon 17, leading to a frameshift [63]. (**C**) Deep intronic variant c.1090+453C>T created a new splice donor sequence, which led to the inclusion of a 77-bp pseudo-exon in the mRNA, causing a frameshift and the introduction of a premature termination codon (PTC) [62]. (**D**) Deep intronic variant c.1091-575A>C created a new splice acceptor sequence, which led to the inclusion of an 85-bp pseudo-exon in the mRNA, causing a frameshift and the introduction of a PTC [62]. (**E**) Synonymous variant c.2274C>T (p.Gly758Gly) caused the truncation of exon 23 by 36 nucleotides, leading to an in-frame deletion of 12 amino acids [62].

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
