# Peer review of "RNA Splicing Defects in Hypertrophic Cardiomyopathy: Implications for Diagnosis and Therapy"

_ijms, 2020, doi:10.3390/ijms21041329_

Round 1

Reviewer 1 Report

The short Review "RNA splicing defects in hypertrophic cardiomyopathy: implications for diagnosis and therapy" by Ribeiro offers a well-structured insight into this field of research. The key components of genetic factors determined hypertrophic cardiomyopathy were discussed, including the all-important points. The review provides a clear understanding of new trends using modern cell biology techniques such siRNA interference etc to diagnose and may treat familiar hypertrophic cardiomyopathy. This review study has summarized in a well written form with illustration, however some issue are need to be clarified.

1) Taken myocardial biopsy and subsequent genetically analysis is a gold standard to detect and confirm familiar hypertrophic cardiomyopathy. However, plasma and circulating levels of exosome, miRNAs and non-coding RNAs expression may also reflect to the various forms of familiar hypertrophic cardiomyopathy. The authors should discuss that in the highlights of potential non-invasive diagnostic approach.

2) The current therapeutic approaches (pharmacological) for the treatment of familiar hypertrophic cardiomyopathy is not elaborated, its worth to summary in brief.

3) Figure 1 and 2 are not presented in current version of the MS. Please provide these two figures.

Author Response

Reply to reviewer #1:

1) Taken myocardial biopsy and subsequent genetically analysis is a gold standard to detect and confirm familiar hypertrophic cardiomyopathy. However, plasma and circulating levels of exosome, miRNAs and non-coding RNAs expression may also reflect to the various forms of familiar hypertrophic cardiomyopathy. The authors should discuss that in the highlights of potential noninvasive diagnostic approach.

R: We thank the reviewer for this suggestion. We have included a new paragraph addressing this topic at the end of the section “RNA therapeutics for HCM”.

2) The current therapeutic approaches (pharmacological) for the treatment of familiar hypertrophic cardiomyopathy is not elaborated, its worth to summary in brief.

R: In the “Introduction”, we included a new paragraph on current therapeutic approaches.

3) Figure 1 and 2 are not presented in current version of the MS. Please provide these two figures.

R: High-resolution versions of all figures have now been submitted.

Reviewer 2 Report

The review by Ribeiro and collogues entitled: “RNA splicing defects in hypertrophic cardiomyopathy: implication for diagnosis and therapy”, aimed to summarized recent discoveries of RNA mis-splicing in hypertrophic cardiomyopathy and provides an overview of research that aim to apply the concept of RNA therapeutics for to hypertrophic cardiomyopathy. It is very well written, clear, and interesting review article while also falling within the scope of the International journal Molecular Sciences.

Only one minor comment: the high images resolution is needed for the figures, in particular, figure 4 and 7.

Author Response

Reply to reviewer #2:

Only one minor comment: the high images resolution is needed for the figures, in particular, figure 4 and 7

R: We thank the reviewer for the positive comments. High-resolution versions of all figures have now been submitted.

Round 2

Reviewer 1 Report

The authors have improved the manuscript accordingly.